# Scaling Cross-Embodiment World Models for Dexterous Manipulation

Zihao He[1*], Bo Ai[1*], Tongzhou Mu[1], Yulin Liu[1], Weikang Wan[1], Jiawei Fu[1],
Yilun Du[2], Henrik I. Christensen[1], and Hao Su[1,3]

[1]University of California, San Diego    [2]Harvard University    [3]Hillbot    *Equal contribution

*Abstract*— **Cross-embodiment learning seeks to build generalist robots that operate across diverse morphologies, but differences in action spaces and kinematics hinder data sharing and policy transfer. This raises a central question: Is there any invariance that allows actions to transfer across embodiments? We conjecture that environment dynamics are embodiment-invariant, and that world models capturing these dynamics can provide a unified interface across embodiments. To learn such a unified world model, the crucial step is to design state and action representations that abstract away embodiment-specific details while preserving control relevance. To this end, we represent different embodiments (e.g., human hands and robot hands) as sets of 3D particles and define actions as particle displacements, creating a shared representation for heterogeneous data and control problems. A graph-based world model is then trained on exploration data from diverse simulated robot hands and real human hands, and integrated with model-based planning for deployment on novel hardware. Experiments on rigid and deformable manipulation tasks reveal three findings: (i) scaling to more training embodiments improves generalization to unseen ones, (ii) co-training on both simulated and real data outperforms training on either alone, and (iii) the learned models enable effective control on robots with varied degrees of freedom. These results establish world models as a promising interface for cross-embodiment dexterous manipulation.**

## I. INTRODUCTION

Cross-embodiment learning underpins the vision of generalist embodied agents that operate across diverse robots despite manufacturing variation, degeneration, or hardware upgrades. Prior progress has shown embodiment-level generalization in locomotion [3] and in manipulation with parallel grippers [4, 9], whereas dexterous manipulation has largely been limited to grasping [17] and in-hand reorientation [**Patel2024GetZero**]. Extending such generalization to broader task domains, such as deformable object manipulation, remains challenging due to complex object dynamics and the need for fine-grained contact control.

Dexterous hands are particularly compelling for cross-embodiment learning because of their anthropomorphic design: their similarity to human hands invites a unified view of learning from *both* human and robot data as a cross-embodiment problem. This raises a central question: what knowledge underlies purposeful action across embodiments with distinct kinematics and control spaces, and how can we encode it so that data from humans and robots become jointly useful? We posit that *world models* [1], mental intuitive physics models learned dynamics models inspired by intuitive physics [15], provide such an interface. The crux lies in state and action design: with an embodiment-agnostic abstraction, heterogeneous datasets can be unified, and the same predictive model can guide control on novel embodiments.

To this end, we represent both human and robot hands as *particles*, with actions defined as particle displacements. A graph-based dynamics model [12, 2, 13, 14, 21] predicts particle motion while exploiting spatial locality and equivariance. We *co-train* on real human–object interaction data and simulated robot–object interaction data. For control, robot joint actions are mapped to particle displacements via forward kinematics, enabling model-predictive planning in the shared state and action space. This abstraction unifies data sharing and control problems across embodiments, avoiding motion retargeting and task-specific expert demonstrations.

We evaluate in simulation and on real hardware. In simulation, we observe an *embodiment-scaling* trend: training on a larger, more diverse set of simulated hands improves generalization to unseen embodiments. In real-world experiments, models trained on human hand transfer directly to two distinct robots, a 6-DoF PSYONIC Ability Hand and a 12-DoF Robot Era XHand, to perform fine-grained deformable manipulation (e.g., dough reshaping). These preliminary results suggest particle-based world models are a viable, generalizable interface for cross-embodiment dexterous manipulation.

## II. METHOD

Our goal is to enable dexterous manipulation skills from and for diverse robotic hands. We formalize the general problem as follows. At each time step $t$, the end effector is in configuration $q_t \in \mathbb{R}^{n_e}$, where $n_e$ is the number of degrees of freedom of embodiment $e$, and the object is in state $s_{obj}$. The world state includes the state of both the robot and the object, $s_t = \langle q_t, s_{obj} \rangle$. The robot takes an action $u_t$, and the world transits to a new state $s_{t+1}$. The objective is to find an action sequence of length $H$, $u_{0:H-1}$, that minimizes a cost function $\mathcal{J}$:

$$u^*_{0:H-1} = \arg \min_{u_{0:H-1} \in \mathcal{U}} \mathcal{J}\big(\mathcal{T}(s_0, u_{0:H-1}), s_g\big), \quad (1)$$

where $\mathcal{T}(s_0, u_{0:H-1})$ is the state reached after applying the sequence to the dynamics, and $s_g$ is the target state.

What is the shared underlying process across different embodiments for these control problems? Our key insight is that the underlying physical interaction process, captured by $\mathcal{T}$, is universal. However, approximating $\mathcal{T}$ is challenging due to the varying dimensions of the robot configuration $q_t \in \mathbb{R}^{n_e}$ and action $u_t \in \mathbb{R}^{n_e}$, which depend on the embodiment $e$, as well as the differences in kinematic and

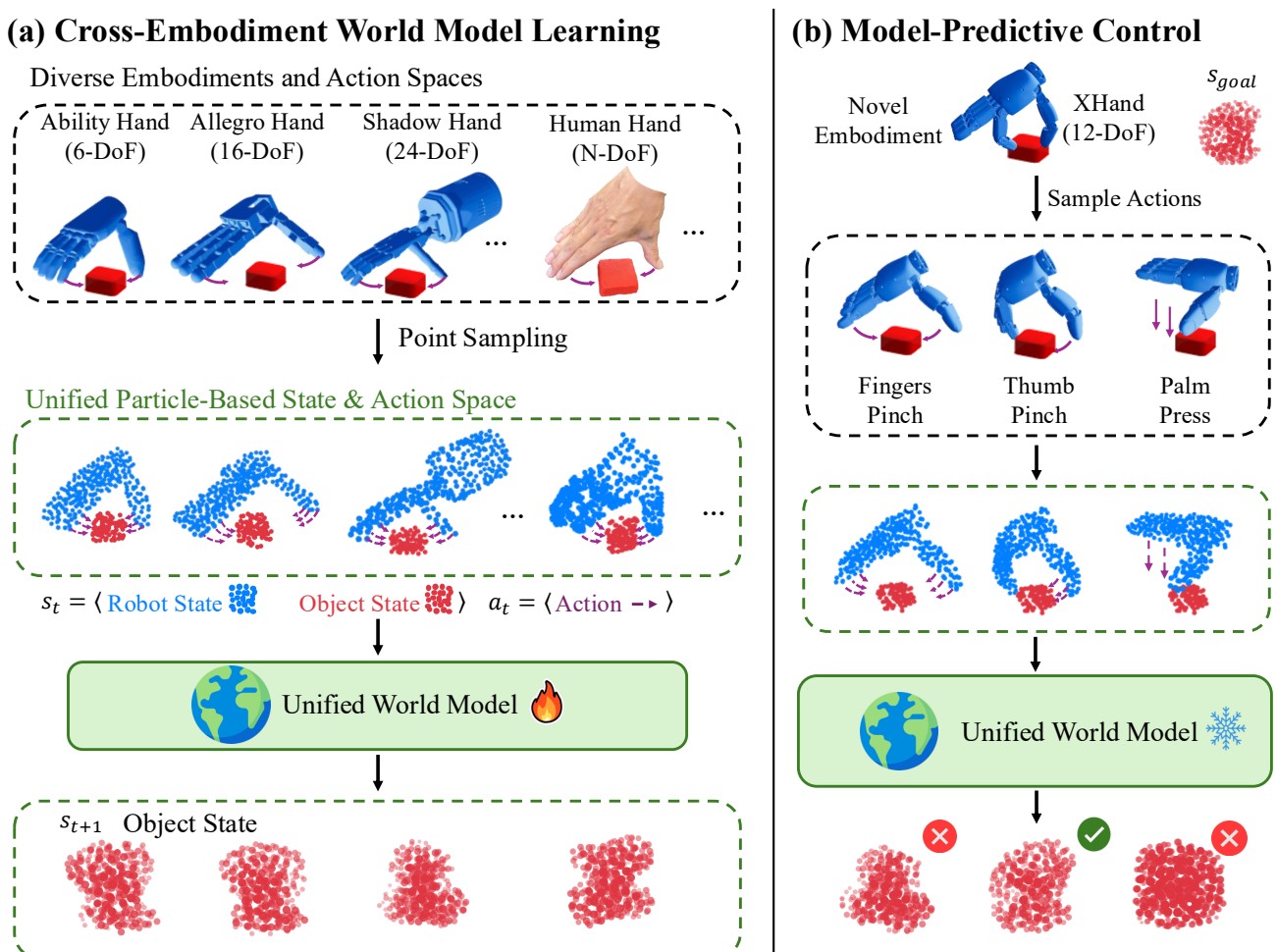

**Fig. 1: Overall framework.** Our key idea is to represent both embodiments and objects as 3D particles, and actions as end-effector particle displacement fields. These state–action abstractions unify data and control across embodiments. (a) We train world models on random interaction data from diverse robot hands in simulation and from human demonstrations in the real world. (b) At deployment, joint action samples are mapped into displacement fields via forward kinematics, rolled out by the world model for prediction, and the optimal trajectory is executed on the target hardware. We show a single-step planning horizon here for simplicity.

geometric structures that shape the environment dynamics. Therefore, we aim to unify state and action representations to learn embodiment-agnostic world models, which hold the potential to scale with cross-embodiment datasets.

We next discuss the high-level framework of cross-embodiment model learning and planning (Section II-A), state estimation (Section II-B), world model architecture (Section II-C), and model-based control (Section II-D).

*A. Cross-Embodiment World Model Learning and Planning*

We define a **particle state and action space** that unifies cross-embodiment data format and control problems. For embodiment $e$, we represent the end-effector at time $t$ by a set of $N_e$ particles, $X_t^{(e)} = \{ x_{i,t}^{(e)} \in \mathbb{R}^3 \}_{i=1}^{N_e}$, the object by $N_o$ particles $X_t^{(o)} = \{ x_{i,t} \in \mathbb{R}^3 \}_{i=1}^{N_o}$, and thus the world state is represented as $X_t = (X_t^{(e)}, X_t^{(o)})$. This is a unified particle-based representation applicable to nearly arbitrary end effector (e.g., multi-fingered hands with different DoFs) and objects (e.g., rigid and deformable objects).

In the particle space, the action can be defined as the end-effector particle displacement field:

$$a_t^P = \Delta X_t^{(e)} = \{ \delta_{i,t} \in \mathbb{R}^3 \}_{i=1}^{N_e},$$

with $X_{t+1}^{(e)} = X_t^{(e)} + \Delta X_t^{(e)}$. This action information can be computed from passive human-object or robot-object interaction data. We can thus train a world model $f$ to approximate the true transition function $\mathcal{T}$ via supervised learning, which predicts the next state given the action:

$$\hat{X}_{t+1} = \hat{f}_\theta(X_t, a_t^P),$$

and

$$\theta^* = \arg\min_\theta \mathbb{E}\left[ \mathcal{L}\left( \hat{f}_\theta(X_t, a_t^P), X_{t+1} \right) \right]. \quad (2)$$

The advantage is that we do not require demonstration data. Random interaction data suffices.

During planning, we obtain particle representations from joint states via forward kinematics (FK). Let $\Phi_e : \mathbb{R}^{n_e} \to (\mathbb{R}^3)^{N_e}$ denote the FK mapping for embodiment $e$. Given the current and next joint states, $q_t$ and $q_{t+1} = q_t + u_t$, the

corresponding particle sets are

$$X_t^{(e)} = \Phi_e(q_t), \qquad X_{t+1}^{(e)} = \Phi_e(q_{t+1}).$$

The shared particle action is then the displacement field

$$a_t^P = X_{t+1}^{(e)} - X_t^{(e)} \in (\mathbb{R}^3)^{N_e}.$$

Planning and learning therefore operate in the embodiment-agnostic spaces $\mathcal{S}^P = (\mathbb{R}^3)^{N_e} \times (\mathbb{R}^3)^{N_o}$ and $\mathcal{A}^P = (\mathbb{R}^3)^{N_e}$. This abstraction enables training on data from diverse embodiments and deployment across different hardware without assumptions about the underlying kinematic structure (e.g., degrees of freedom). The only requirement is a forward kinematics model to map joint actions into the particle action space, which is a mild assumption since robot models are typically available at deployment. We illustrate the overall framework in Figure 1.

### B. Perception Module

The perception module performs state estimation for data collection and deployment. We use a multi-view camera setup [2, 12, 13, 16]. Cameras are placed at fixed positions around the scene so that each captures the interaction from a different viewpoint, ensuring comprehensive coverage.

For real-world human data collection, we reconstruct hand meshes from the multi-view images using POEM-v2 [20], and sample particles with farthest point sampling (FPS). For object perception, we fuse multi-view point clouds, perform Poisson surface reconstruction to obtain a smooth surface [8], and apply FPS. In both cases, FPS allows us to obtain particles that preserve the full geometry of the meshes.

### C. World Model Architecture

We consider adopting graph neural networks (GNNs) as our world model architecture, as the locality and equivariance are useful inductive biases [7] that allow the learned model to generalize to objects and hands with different shapes. We use DPI-Net [10], a GNN that models local particle interactions through message passing and captures global effects via multi-step hierarchical propagation.

The particle-based graph network incorporates strong inductive biases. Spatial locality is enforced by restricting message passing to local neighborhoods. Equivariance is achieved through relative coordinates and shared update functions, ensuring invariance to global translations, rotations, and particle permutations. These properties support generalization across embodiments.

### D. Model-Based Planning

Inspired by the insight that human hand motions lie in low-dimensional manifolds of the full configuration space [6], we design action spaces for efficient planning, and use the cross-entropy method for model-predictive control (MPC).

For the *Object Pushing* task, we constrain pushing to a fixed $x$–$y$ plane. Global translations are sampled as random motion noise in the end-effector frame, while the number of fingers making contact with the box is randomly selected.

For the *Plasticine Reshaping* task, we define a low-dimensional action parameterization with three motion primitives: (i) `FingersPinch`, involving rotation about the $z$-axis and relative motion between the index finger and thumb; (ii) `PalmPress`, characterized by rotation about the $z$-axis and translation along the $z$-axis; (iii) `ThumbPinch`, composed of rotation about the $z$-axis and actuation of thumb-specific degrees of freedom.

For model-based control, we sample control sequences $\{u_t\}_{t=0}^{H-1}$ from the robot hand's action space defined by each primitive. These are mapped to particles in the shared state space through forward kinematics, rolled out with the learned world model, and evaluated using the cost function. The target is specified as a point cloud, and the cost measures the similarity between the predicted final state and the target using distances such as CD or EMD.

## III. Experiments

In this section, we study the following questions:

**Q1.** Does cross-embodiment training of the world model improve generalization on unseen embodiments?
**Q2.** What is the co-training recipe to leverage simulation and real-world data?
**Q3.** Does the learned dynamics model enable effective planning for dexterous manipulation?

### A. Experimental Setup

**Task setup.** We consider two representative dexterous manipulation tasks: non-prehensile rigid object pushing [2, 5] and deformable object reshaping [2, 12, 13]. In *Object Pushing*, the goal is to reorient a box to a target orientation. In *Plasticine Reshaping*, the goal is to mold plasticine into a target shape specified by a point cloud. Both tasks require precise contact control and reasoning about object dynamics.

**Simulation setup.** We simulate six dexterous hands representative of commonly used multi-fingered designs: Ability Hand (6-DoF), Allegro Hand (16-DoF), XHand (12-DoF), Leap Hand (16-DoF) [11], Shadow Hand (24-DoF), and a URDF variant of the Shadow Hand without its forearm (24-DoF). For the rigid-body task (*Object Pushing*), we use SAPIEN [18] for data collection. For deformable object manipulation, we use the Rewarped simulation platform [19]. We collect 100 trajectories per task, where the robots perform random actions in the predefined action space.

**Real-world setup.** Our hardware platform consists of a 7-DoF XArm robot equipped with an Ability Hand and an XHand. Four Intel RealSense cameras provide multi-view perception. The system is controlled via a workstation with an NVIDIA RTX 4090 GPU. For human demonstration data, we collect 30 minutes of demonstrations for `ThumbPinch`, `FingersPinch`, and `PalmPress` each.

### B. Evaluating Cross-Embodiment World Model Learning

We systematically evaluate how the number of training embodiments influences generalization to unseen embodiments. For each target hand, we hold it out and train on $x$ other hands, enumerating all $\binom{N}{x}$ subsets from $N = 6$

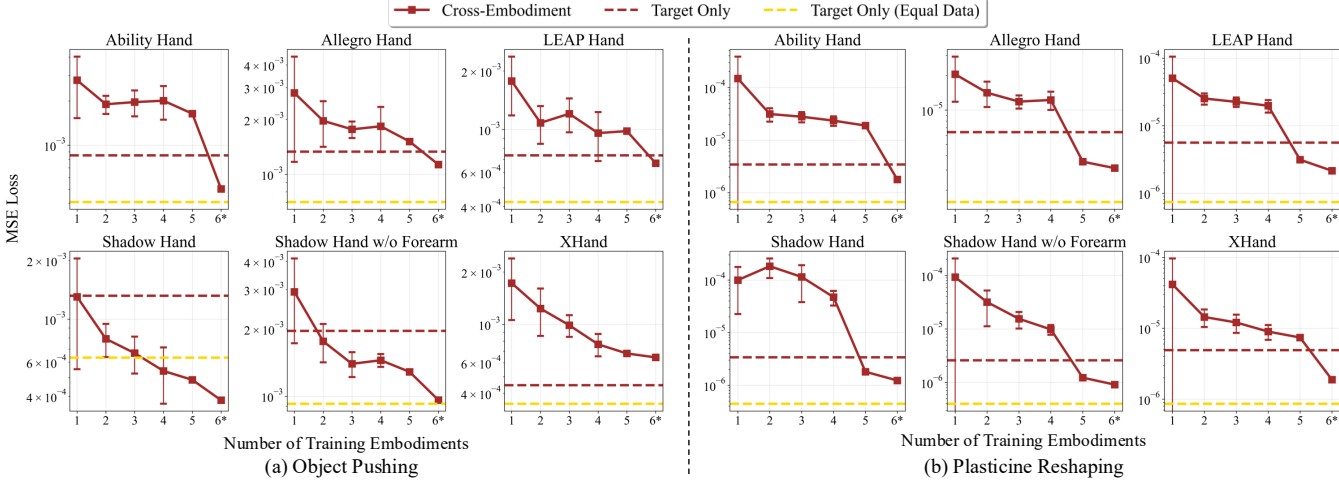

**Fig. 2: Scaling trends in cross-embodiment world model learning.** For each target hand, models are trained on subsets of the remaining hands of varying sizes. All subset combinations at a given size are enumerated (e.g., $\binom{5}{2}$ for size 2), and the mean performance with 95% confidence intervals is reported. Dashed lines indicate models directly trained on the target embodiment.

total hands. The mean squared error (MSE) on the unseen hand serves as the generalization metric. In addition, the case $x = 6$ corresponds to training on all hands, including the target, and provides a reference for the upper bound of cross-embodiment learning in the current data regime. Results are shown in Figure 2.

**Key observations.** We make the following observations:

- *Embodiment scaling law [3]:* Prediction error decreases as more embodiments are included, and variance across subsets shrinks, indicating more stable models with broader embodiment diversity.
- *Zero-shot strength at $x$=5:* With five training embodiments (no target data), performance often approaches or surpasses training directly on the target hand. This shows that diverse cross-embodiment data can substitute for target-specific data when deploying to a new hand. This opens up the possibility of building cross-embodiment generalist world models that can broadly zero-shot transfer to novel ones via large-scale cross-embodiment training.
- *Benefit of co-training at $x$=6*:* Even when target data is available, adding the other embodiments yields further gains over target-only training. Our proposed state and action representations unify data from heterogeneous embodiments and make such co-training possible.

**Task-specific differences.** Errors are generally lower for deformable reshaping, as deformations are spatially localized, whereas rigid-body rotations move particles over a much larger scale. At the same time, the scaling effect is more pronounced in deformable manipulation. We hypothesize this is because deformable tasks involve larger contact surfaces, making end-effector geometry more influential. Exposure to diverse embodiments therefore provides richer coverage of contact geometries and interaction patterns, which aids generalization.

**Embodiment-specific trends.** Certain hands (e.g., Shadow Hand, Leap Hand) show sharp improvements

when scaling from 4 to 5 training embodiments, whereas smaller hands (e.g., Ability Hand) achieve competitive performance earlier. We hypothesize that this effect is linked to graph density in the particle–graph representation used by our GNN-based world model. Smaller hands have fewer degrees of freedom but a more compact geometry, which results in denser particle connections under the radius-graph construction. This denser connectivity provides richer local message passing and allows the GNN to propagate interaction information more effectively, even when trained on fewer embodiments. By contrast, larger hands span a larger spatial extent, yielding sparser graphs where local neighborhoods capture fewer interactions. In such cases, broader embodiment diversity is needed to expose the model to sufficient variations in contact patterns and fill in the missing structural information.

### C. Co-Training Recipe

Having established positive embodiment scaling in simulation, we next study how to leverage simulation data for real-world learning. Simulation offers uniform sensing and abundant interactions, but models trained purely in simulation can overfit to simulator-specific artifacts such as contact or material mismatches. Conversely, real-world human data avoids the reality gap but introduces an embodiment gap relative to robot hands. We hypothesize that co-training on both domains may combine their complementary strengths, when the signals from each are balanced appropriately.

We train models with different mixtures of simulation and real-world data, and evaluate them on held-out human data (Figure 4). Simulation-only training yields the highest prediction error, highlighting the sim-to-real gap. Human-only training provides a stronger baseline, and mixing simulation with human data further reduces error when the ratio is well balanced. Notably, a 1:1 ratio performs best across tasks, suggesting that simulation data can act as a useful regularizer for human data rather than a substitute. We emphasize, however, that these evaluations are on human data only, since

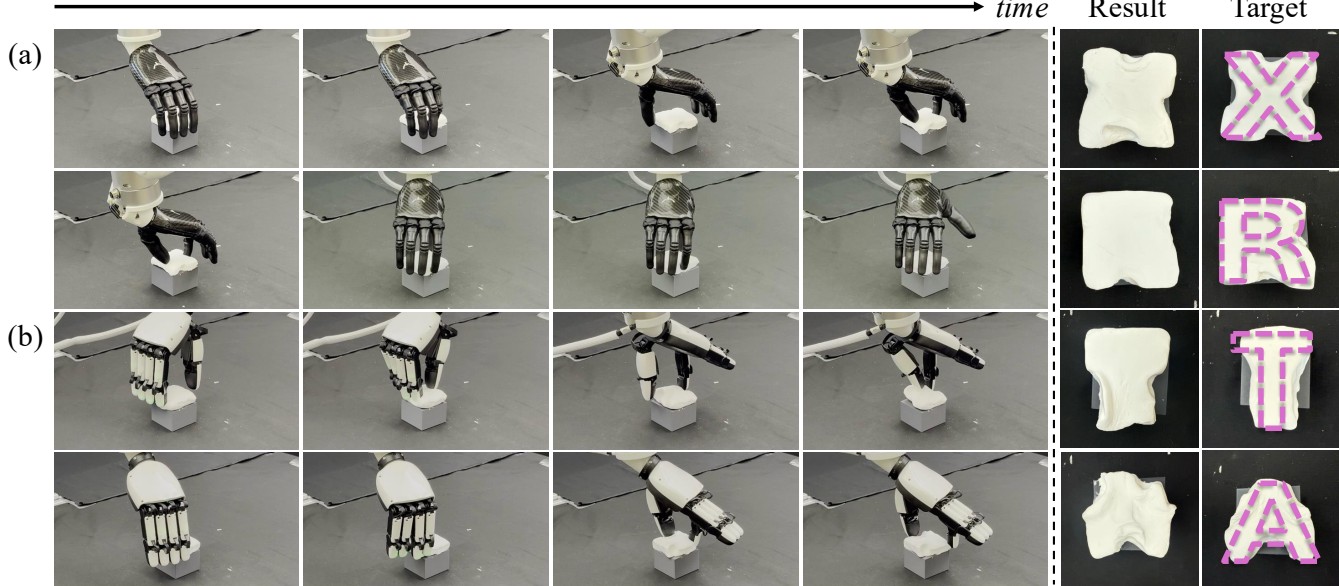

**Fig. 3: Qualitative results of cross-embodiment deployment**. (a) Ability Hand (6-DoF) and (b) XHand (12-DoF) utilize the same particle-space dynamics model learned from human demonstration. For each trial, the hand successfully reshapes the deformable clay toward the target shape using a combination of `FingersPinch`, `PalmPress`, and `ThumbPinch` skills.

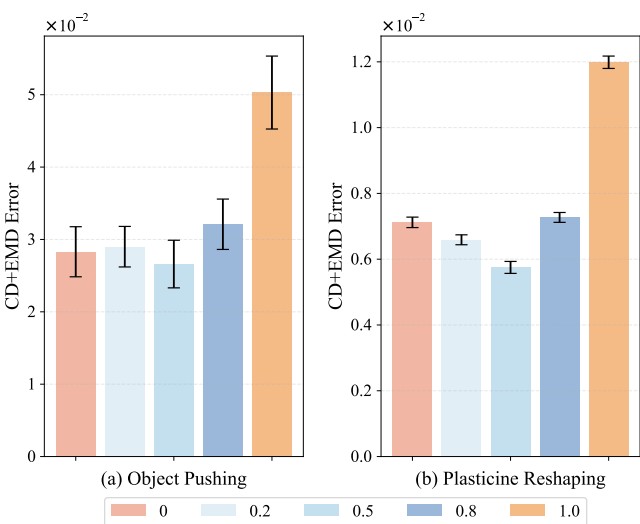

**Fig. 4: Evaluating training recipes for bridging simulation and real.** We compare co-training with different mixtures of simulation and real-world data. Legend values indicate the amount of simulation data relative to a fixed quantity of real human data. The y-axis shows prediction error on held-out human interactions, with error bars denoting 95% confidence intervals.

| Hand | Method | CD $\times 10^{-3}$ $\downarrow$ | EMD $\times 10^{-3}$ $\downarrow$ |
|---|---|---|---|
| Ability | **Co-train** | **6.95 $\pm$ 0.10** | **4.92 $\pm$ 0.13** |
| | Only Human | 7.15 $\pm$ 0.12 | 5.23 $\pm$ 0.17 |
| XHand | **Co-train** | **6.85 $\pm$ 0.13** | **4.78 $\pm$ 0.15** |
| | Only Human | 7.22 $\pm$ 0.19 | 5.18 $\pm$ 0.22 |

**TABLE I:** Performance comparison of co-training (human + 6 simulated robot hands) vs. training on only human data, evaluated on Ability Hand and XHand. Reported values are mean $\pm$ 95% confidence interval. Lower is better.

Quantitative results are reported in Table I. Both models leverage the unified state and action space to operate seamlessly across embodiments. The human-only model achieves zero-shot transfer to novel robot hands, but its performance is lower than that of the co-training model. Qualitative results of the co-training model are shown in Figure 3. Both the *(a)* Ability Hand and *(b)* XHand successfully reshape clay into target letters by composing the three learned skills, `ThumbPinch`, `FingersPinch`, and `PalmPress`, to carve, spread, and compress. Despite their kinematic differences, the same particle-based dynamics model enables model-predictive planning on both hands without fine-tuning, demonstrating effective cross-embodiment deployment.

## IV. CONCLUSION

This work shows that a unified state and action representation, combined with world model learning and model-predictive control, can enable dexterous skills to transfer across diverse physical embodiments. While preliminary, our results highlight world models as a promising abstraction for unifying heterogeneous data and control, and we hope this direction will inspire further exploration toward generalist cross-embodiment policies across broader tasks and morphologies.

the target embodiment data is not available; the values only serve as approximations of the target domain. We will further verify the benefits of co-training through system deployment.

### D. Evaluating Model-Based Control

For real-world deployment, we focus on *Plasticine Reshaping*, which is more challenging due to complex contact dynamics. We compare two models: one trained on human data only and the best-performing co-training model. Each model is evaluated across four target shapes ("X", "R", "T", "A"), with five trials per shape, for a total of 20 runs.

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
