# OpenReview forum: "Scaling Cross-Embodiment World Models for Dexterous Manipulation"
_robot-learning.org/CoRL/2025/Workshop/Dexterous_Manipulation — CoRL 2025 Workshop Dexterous Manipulation Spotlight_

### Official Review · Reviewer_Zei9 · 2025-09-10
**Neural Model Learning for Cross-Embodied DexHand for Deformable Object Manipulation**

**Rating:** 7
**Confidence:** 4

**Review:**

Summary: The paper proposes a novel way to learn a point-based dynamics model for deformable objects and dexterous hand. Moreover, it supports generalization to dexterous hand of different morphologies. To achieve this, the paper proposes a way to represent the hand for all kinds of grippers and use a GNN to model the dynamic of object points and the hand points. The model is trained on both simulation data and real-world data collected with human hand. Experiments show the model does allow relative accurate cross-embodiment dynamic model learning and shows result of using the model for dexterous manipulation task of deformable objects.
Strength: The method is novel and the way of unifying the dex hand repr into the dynamics model in interesting. The experiment is thorough and complete.
Weakness: It primarily focuses on simple skills which is pre-defined. Extending the pipeline to more complex skills require collecting more data. The paper has not discussed the limitation / possibility of applying the framework on more diverse dex hand motions. The qualitative results showed in real world is not in good shape. Is it possible to train RL with the neural dynamics model?
Q&A: Why does the author need real-world data and simulation data. What is the performance if one only uses one source of data?

---

### Official Review · Reviewer_L2pg · 2025-09-10

**Rating:** 8
**Confidence:** 4

**Review:**

The core idea is to represent both human and robot hands as a set of particles, then model how those particles move when manipulating deformable objects like dough. This way, you get a unified “language” or representation that doesn’t care about the actual shape or complexity of the hand.

The particle-based representation makes it easy to share data between humans and robots without complicated retargeting or expert demonstrations, which is often a pain point.

Training on multiple robot hands improves generalization to unseen hands, which is a nice confirmation that diversity helps learning.

The real-world experiments are solid — they actually tested this on two very different robot hands reshaping dough.

Model-predictive control integrated with the learned world model lets the robots plan actions in this shared space, enabling them to perform complex manipulation skills like pinching and pressing in a robust way.

One thing I’d be curious about is how well this approach scales with more complex skills/tasks or objects beyond dough. Also, how sensitive is the system to perception errors or occlusions in more cluttered environments?

---

### Decision · Program_Chairs · 2025-09-18

Accept (Spotlight)